# Disassembly of Amyloid Fibril with Infrared Free Electron Laser

**DOI:** 10.3390/ijms24043686

**Published:** 2023-02-12

**Authors:** Takayasu Kawasaki, Koichi Tsukiyama, Phuong H. Nguyen

**Affiliations:** 1Accelerator Laboratory, High Energy Accelerator Research Organization, 1-1 Oho, Tsukuba 305-0801, Japan; 2Department of Chemistry, Faculty of Science Division I, Tokyo University of Science, 1-3 Kagurazaka, Tokyo 184-8501, Japan; 3CNRS, UPR 9080, Laboratoire de Biochimie Théorique, Institut de Biologie Physico-Chimique, Fondation Edmond de Rothschild, Université Paris Cité, 13 Rue Pierre et Marie Curie, 75005 Paris, France

**Keywords:** amyloid fibril, infrared free electron laser, amide I, β-sheet, α-helix, vibrational excitation, disassembly

## Abstract

Amyloid fibril causes serious amyloidosis such as neurodegenerative diseases. The structure is composed of rigid β-sheet stacking conformation which makes it hard to disassemble the fibril state without denaturants. Infrared free electron laser (IR-FEL) is an intense picosecond pulsed laser that is oscillated through a linear accelerator, and the oscillation wavelengths are tunable from 3 μm to 100 μm. Many biological and organic compounds can be structurally altered by the mode-selective vibrational excitations due to the wavelength variability and the high-power oscillation energy (10–50 mJ/cm^2^). We have found that several different kinds of amyloid fibrils in amino acid sequences were commonly disassembled by the irradiation tuned to amide I (6.1–6.2 μm) where the abundance of β-sheet decreased while that of α-helix increased by the vibrational excitation of amide bonds. In this review, we would like to introduce the IR-FEL oscillation system briefly and describe combination studies of experiments and molecular dynamics simulations on disassembling amyloid fibrils of a short peptide (GNNQQNY) from yeast prion and 11-residue peptide (NFLNCYVSGFH) from β2-microglobulin as representative models. Finally, possible applications of IR-FEL for amyloid research can be proposed as a future outlook.

## 1. Introduction

Protein structures are generally constructed by a number of peptide fragments such as α-helices and β-sheets, and many functional proteins have globular shapes including α-helices mainly under physiological conditions. In contrast, β-sheet-rich conformation can often be dominant over the α-helix or other conformations when amino acid mutations occur and protein concentrations are increased locally in cells [1,2,3]. Amyloid fibrils are such a protein assembly and are deposited into various tissues, resulting in onset of amyloidosis [4,5,6]. A part of the representative amyloidosis and the causal amyloid proteins are listed in Table 1. About 40 kinds of proteins cause amyloidosis and can be mainly divided into 2 categories [7]: the first group is related to neurodegenerative diseases which includes Aβ [8,9], tau protein [10], polyglutamine [11], transthyretin [12], prion protein [13], S100 protein [14], and α-synuclein [15]. The second group contains lysozyme [16], calcitonin [17], insulin [18], and β2-microglobulin [19]. Interestingly, the wavenumbers of the amide I of their fibril states are around 1610–1640 cm^−1^, while those of globular proteins are usually around 1640–1660 cm^−1^ [20]. The difference in the wavenumber is caused by a cross β-sheet stacking formation which is common in many amyloid proteins, although how the fibril structure is associated with the cell toxicity remains unclear.

The structure of the amyloid fibril is constructed by self-assembled β-sheet chains: the diameter of a chain varies from 1 to 10 nm and its length is usually several 100 nano-meters (Figure 1). Both fibrils from hen egg-white lysozyme and human insulin are formed under acidic conditions, and those fibrils seem to twist like thin strings [21,22]. It is considered natural that deconstruction of the fibril structure can halt aggregation of the amyloid proteins and interfere with the disease worsening, which should become an efficient treatment. However, the fibril structure is highly hydrophobic and formed by a hydrogen bonding network [23,24,25]. In addition, the amyloid fibril is strongly deposited into the biological tissues, so it is difficult to dissociate the β-sheet stacking structure under physiological conditions without denaturants and organic solvents such as dimethyl sulfoxide [26,27].

Infrared free electron laser (IR-FEL) is an intense picosecond (ps) pulsed laser (Figure 2) [28,29,30], and the mechanism of IR-FEL oscillation is briefly as follows: An electron beam is generated by a high radio-frequency (RF) electron gun (2856 MHz) and injected into an undulator (a periodic magnetic field) through a linear accelerator. The maximum acceleration energy reaches 40 MeV, and the electron beam is bent periodically to generate the synchrotron radiation (SR) in the undulator. The SR interacts with the electron beam and produces the IR-FEL. The time structure exhibits double pulse domains where several thousands of micropulses having a 1–2 ps duration are bunched with a 350 ps interval in one macropulse. The macropulse has a 2 μs duration and is oscillated at 5 Hz, and its pulse energy reaches 10 to 20 mJ. A fascinating characteristic of IR-FEL for users is the capacity to excite the various vibrational modes independently due to the wavelength tunability from near to far infrared regions (3–100 μm), which can lead to alteration of the molecular structure of various compounds in biological, gas-phase, and solid materials. Presently, IR-FELs are open at many synchrotron-radiation facilities all over the world since the optical emission was first observed at a laboratory level more than 50 years ago [31]. Various original experimental studies using IR-FELs are reported as follows: ablation of biological tissues [32,33,34], alteration of biological functions [35,36], degradation of protein–metal complexes [37,38], optical and spectroscopic imaging [39,40,41], structural dynamics of biomolecules [42,43], chemical reactions in the gas phase [44,45,46], decomposition of biopolymers [47,48,49], nonlinear-optical physics [50], and observation of terahertz radiation [51].

We discovered that IR-FEL tuned to amide I can disassemble the β-sheet stacking structures of amyloid fibrils into their non-fibril states containing α-helix conformations (Figure 3) [21,22,52,53,54,55,56,57,58,59,60,61]. In almost all cases, the silk-like fibrils are converted to tiny particles after laser irradiation. Various fibrils from lysozyme (129 residues), insulin (51 residues), DFNKF (5 residues), polyglutamine (69 residues), Aβ (42 residues), GNNQQNY (7 residues), and β2-microglobulin peptide (11 residues) can be dissociated to their non-fibril states, although the disassembly patterns are slightly various depending on the amino acid sequences. In addition, those experimental results were arguably evidenced by molecular dynamic simulations [62,63,64,65,66]. Interestingly, the conversion from β-sheets to α-helices is observed in all types of amyloid proteins.

It can be estimated that the dissociation of amyloid fibrils can proceed within several hundred picoseconds (Figure 4), which was theoretically verified by a non-equilibrium simulation study as described later [58,59]. A thermal confinement theory can be applied to this laser-mediated dissociation process: the fibril dissociation is finished before the diffusion of the vibrational excitation energy because an energy scape to the outer environment requires several nanoseconds [67]. The fibril state is thermodynamically more stable than the native state [68], and so it can be inferred that the vibrational excitation energy from IR-FEL can overcome the energy barrier before the thermal diffusion, which can produce the disassembled peptides.

The applications of IR-FEL in the dissociation of amyloid fibrils have been demonstrated for various systems. From the theoretical side, molecular dynamics simulations have been performed for various fibril models, including Aβ_17-42_ and HET-s [62], GNNQQNY fibrils [59], pure polyglutamine and pure polyasparagine fibrils [69], and tetrameric Aβ_1-42_ β-barrel in neuronal membrane [64]. From the experimental side, we have investigated the conformational changes of several fibrils from lysozyme [21,58], insulin [22], DFNKF [52,53,57], polyglutamine [54], Aβ_1-42_ [56], GNNQQNY [59], and β2-microglobulin [60] by using infrared microspectroscopy, electron microscopy, and the fibril-binding fluorescent reagents. In this review, we focus on two representative amyloid fibril systems which have been investigated by joint experimental/theoretical studies [59,60].

## 2. Infrared-Laser Induced Dissociation of GNNQQNY Peptide Fibril

Prion is known to cause several neurodegenerative diseases and Creutzfeldt–Jakob disease in mammals [70]. Prion also exists in bacteria, and a part peptide of Sup35, GNNQQNY, is self-assembled to form amyloid fibrils [71,72,73]. This peptide fibril was subjected to the IR-FEL irradiation and analyzed by both experiments and non-equilibrium molecular dynamics simulation [59].

### 2.1. Experimental Results

The laser beam from the IR-FEL oscillation system was introduced to the surface of the fibril sample, and the beam diameter was focused to about 0.3–0.4 mm using a barium fluoride lens. The pulse energy was 9−10 mJ, and the local destruction of the sample surface can be seen after 10 macropulses where the duration of one macropulse is 2 μs [74]. Below 5 mJ, the irradiation effect on the fibril dissociation was weak. In this study, the number of irradiation pulses was set to 900 macropulses (=3 min) to induce full dissociation of the fibril state.

FT-IR spectrum of the non-fibrous conformation of the peptide (Pre-fibril) exhibits a dominant peak at 1650 cm^−1^ and a shoulder peak at 1631 cm^−1^ (≈6.13 μm) at the amide I region (Figure 5a). After fibrillation (mature-Fibril), the absorbance of the latter peak increases, which corresponds to the C=O stretching mode in the β-sheet chain, and the experimental wavelength of IR-FEL was set to 6.13 μm. A non-specific wavelength (2000 cm^−1^ = 5.0 μm) was also tested as a reference. Figure 5b shows the thioflavin T (ThT) fluorescence intensity of the fibril before and after irradiations at the amide-I resonant and the non-resonant frequencies. The irradiation at 6.13 μm caused a decrease of nearly 85% of the ThT signal, which indicates a substantial decrease in the fibrous conformation, while the irradiation at 5.0 μm hardly changed the signal, which indicates the fibril state was maintained.

In the SEM images (Figure 6a), straw-like filaments (left) markedly changed into small fragments (right) by the irradiation at 1631 cm^−1^. Wide-angle X-ray scattering (WAXS) measurement was performed to examine the conformation of the peptide (Figure 6b). In the Pre-fibril state (blue), one main broad peak was detected at q = 13.25–13.5 nm^−1^, which indicates random peptides in the oligomeric state. On the contrary, a strong peak and a weaker peak were observed in the fibril state at 12.8 nm^−1^ and 13.5 nm^−1^, respectively (red). The former peak means an intermolecular Cα−Cα distance (d) of 0.49 nm in the cross-β sheets. After IR-FEL irradiation at 1631 cm^−1^, this peak was decreased and the broad peak near q = 13.5 nm^−1^ remained, which shows complete dissociation of the fibril (green).

### 2.2. Simulation Results

We performed nonequilibrium molecular dynamics (NEMD) simulation to examine the fibril dissociation process with the IR-FEL irradiation. Based on a calculation of the IR spectrum of 100 structures and a statistical selection from the equilibrium trajectory, the amide-I band around 1675 cm^−1^ was dominantly identified as the C=O stretching vibration in the cross-β sheets, and a band around 1690 cm^−1^ was assigned as the C=O stretching vibrations in the other conformations. Interestingly, it was found that the fibril state was deformed after one single laser pulse at 1675 cm^−1^ by calculations of root-mean-square-deviation (RMSD), number of intermolecular hydrogen bonds, and radius of gyration. This evidenced that the laser irradiation resonant with the C=O stretching vibration in the cross-β sheets causes disruption of the fibril. To examine the effect of the laser intensity, we performed 100 simulations with three laser intensities, E_0_ = 1.5 V/nm, 2.0 V/nm, and 2.5 V/nm at 1675 cm^−1^, and one laser intensity E_0_ = 2 V/nm at 2000 cm^−1^ (Figure 7a). The fibrous assembly was fully dissociated after 500 ps with E_0_ = 2.0 V/nm and 2.5 V/nm while the fibril was partially dissociated in the case of E_0_ = 1.5 V/nm and was little changed at 2000 cm^−1^ even with the high laser intensity E_0_ = 2.0 V/nm.

To associate the molecular simulation with the experiment, we calculated the WAXS spectra (Figure 7b) and the secondary structures of the peptide (Figure 7c) with E_0_ = 2 V/nm at 1675 cm^−1^. A peak at 13.3 nm^−1^ remained within the first 20 ps in the WAXS spectrum which is close to the peak at 12.8 nm^−1^ in the experimental spectrum (Figure 6b). The peak was shifted to 11.5 nm^−1^ at 24 ps and almost disappeared after 50 ps, which means destabilization of the fibrils. The population of secondary conformation was calculated based on the peak intensity at amide-I frequency: 1625–1640 cm^−1^ for β-sheet, 1650–1655 cm^−1^ for α-helix, 1645–1650 cm^−1^ for other, and 1655–1675 cm^−1^ for turn. Before the IR-FEL irradiation, the populations of β-sheet, random coil, turn, and helix were 68%, 18%, 14%, and 0%, respectively (Figure 7c, left). After the IR-FEL irradiation (right), these populations changed into 11%, 30%, 31%, and 28% in the experiment, which indicates that the fibril was converted to non-fibrous conformation with dominant random coil/turn structures. In the molecular simulation using E_0_ = 2 V/nm, β-sheet, coil, turn, and helix are 5%, 40%, 40%, and 15%, respectively. These results are almost compatible with the experimental data.

To visualize the time evolution at the peptide sequence level, intermolecular hydrogen-bond mapping was created using E_0_ = 2 V/nm with ω = 1675 cm^−1^ as the simulation parameters (Figure 8). Before the laser irradiation (0 ps), five strong hydrogen bonds at N−N, Q−N, N−Q, Q−Q, and N−Y can be observed, which refers to the parallel β-sheets formation. After the laser irradiation, the number of hydrogen-bonds continuously decreased from 100−140 to 20−40 after 50 ps, which indicates that the laser can enhance the separation between peptide chains.

## 3. Disassembly of β2-Microglobulin Peptide Fibril

β2-Microglobulin (β2M) is a protein composed of about one hundred amino acids and is a causal factor for serious amyloidosis in kidney dialysis patients [75]. Especially, the peptide fragment (NFLNCYVSGFH) tends to form a self-assembly under physiological conditions [76]. The peptide fibril was subjected to the IR-FEL irradiation, and the conformation was analyzed by infrared microspectroscopy (IRM) and equilibrium molecular dynamics simulation [60].

### 3.1. Experimental Results

Three wavelengths (6.1 μm (νC=O, amide I), 6.5 μm (δN-H, amide II), and 5.0 μm (non-absorption wavelength)) were selected for the irradiation experiments. In the IRM spectra (Figure 9a), a strong peak at 6.15 μm (1624 cm^−1^) and a broad peak at about 6.0 μm (1667 cm^−1^) were observed (blue, non-irradiation). The former corresponds to the β-sheet and the latter influenced amide bonds associating with other conformations containing α-helix. The laser irradiation tuned to 6.1 μm with an energy fluence of 50 mJ/cm^2^ reduced the peak intensity from the β-sheet (red) more than those tuned to 6.5 μm (violet) and 5.0 μm (green). The peptide conformations were markedly changed when the energy fluence was set to more than 40 mJ/cm^2^ (Figure 9b,c).

### 3.2. Simulation Results

Equilibrium molecular dynamics simulation using GROMACS (version 2019.6) was performed to investigate the conversion process from the β-sheet to α-helix in the 11-residue peptide (Figure 10a). The converged RMSD value of the β-sheet is higher than that of the α-helix, and the fluctuation of the β-sheet was larger than that of the α-helix (Figure 10b). Time evolution analysis suggested that the β-sheet conformation was disordered while the α-helix was stable after the peptide fibril was irradiated by laser irradiation (Figure 3 in [60]).

Hydrogen bonds analysis highlighted a critical amino acid for the conversion from β-sheet to α-helix of the peptide (Figure 11a). The probability of hydrogen bonds at N4 was remarkably decreased from 70% in the β-sheet to 10% in the α-helix. A 3D stick model showed that L3 and C5 are close to N4 and the hydrogen bond distances between N4 and those amino acids are elongated from 2 Å in the β-sheet to 12–14 Å in the α-helix (Figure 11b). Therefore, it can be estimated that the hydrogen bonds associated with N4 can be dissociated by the vibrational excitation induced by the amide-I specific irradiation.

Involvement of water molecules can also be suggested to induce the dissociation of the fibril [77]. NEMD simulation of the Aβ peptide fibril showed that water molecules sometimes enter the space between the β-sheet chains and prevent the reformation of the intermolecular hydrogen bond if the peptide fibril is irradiated by the IR-FEL. Therefore, the water molecules disturb the formation of hydrogen bonds and allow the disassembly of the fibril during the laser irradiation (Figure 12).

## 4. Future Application of IR-FEL in Amyloid Research

The above results showed that the combination of experiments and molecular dynamics simulations gives us a clear image of the disassembly process of amyloid peptides by the IR-FEL, and we can expect that the use of intense infrared lasers will open a way for the reduction of various types of amyloid aggregates in future. Regarding the therapeutic strategies for Alzheimer’s disease (AD), non-pharmaceutical approaches are now highly desirable although several antibodies against Aβ and inactivators of γ-secretase have been tested as pharmaceutical drugs [78,79,80]. In recent years, it has been suggested that the binding of Aβ to cell membranes is a critical step for the onset of AD pathogenicity, and the hydrophobic interactions of cholesterol molecules with Aβ play important roles in β-sheet oligomerization [64,81,82,83,84]. Therefore, it can be supposed that targeting the binding of cholesterol in the lipid membrane to the Aβ fibril may affect the progression of the pathogenicity. The IR-FEL can decompose the cholesterol in blood vessels by tuning the irradiation wavelength to the ester bonds [85]. It can be supposed that the Aβ binding to the lipid membrane can be perturbed by excluding the cholesterol by laser irradiation. In the future, a therapeutic approach based on this concept may lead to a novel strategy for the treatment of AD.

Regarding the physical engineering techniques, it has been so far reported that low-level near-infrared laser can restore functions of mitochondria and induce transcription factors [86]. In addition, the neuronal functions are recovered and the Aβ plaque can be decreased by laser treatment in model animals [87]. It is also reported that a radiofrequency electromagnetic field (1950 MHz) can lead to the improvement of AD-like symptoms in transgenic mice [88]. However, it has not been proven whether those external radiations would induce dissociation of Aβ aggregates by destruction of β-sheet conformation. Therefore, dissociation of amyloid fibrils by using IR-FEL opens up a novel promising approach for treating AD.

However, there remain two important subjects that we should solve towards development of amyloidosis therapy: (1) Although it has been recognized that the low-molecular weight oligomers possess cellular toxicity compared with the integrated fibrils [66,83,89,90], the ability of the IR-FEL to degrade those oligomeric states has not yet been elucidated at the present stage. (2) What about the effect of the laser irradiation on cellular tissues? Regarding the laser effect on the normal tissues, one hypothesis was proposed using lysozyme protein from hen egg-white as a model [58]. We found that the protein conformation was hardly affected on the synchrotron-radiation CD spectrum although the enzymatic hydrolysis activity against the bacterial cell wall was a little decreased by the IR-FEL irradiation at the amide-I band. This is further supported by our molecular dynamics simulations which show that Aβ fibrils are completely dissociated while DNA molecules and globular proteins are hardly affected [62]. Accordingly, IR-FEL can lead to dissociate the fibril structure of amyloids with little damage of the other biomolecules. Nonetheless, the investigation on the IR-FEL irradiation to the mammalian body is a future subject.

On the contrary, it is known that amyloid formation is a so-called “intrinsically disordered” status [91,92]. The intrinsically disordered proteins play important roles such as gene regulators in cells. There is a possibility that the cell function related with the functional amyloids can be altered by the physical engineering technique using IR-FEL. The study of the regulation of the biological functions related with intrinsically disordered peptides should be a future research project.

One more issue that we should mention about the application of IR-FEL in amyloid research is that amyloid fibers have a potential application in functional biomaterials due to the rigidness and the regularity in the fiber structure, similar to cellulose fibers [93,94,95,96,97,98]. For example, the fiber format can be applied to adhesive sheets for cell cultivation and a nano-carrier for a drug-delivery system. In the former case, infrared laser irradiation can be applied to remove the fibrous hydrogel from tissues, and in the latter case, the drug in the vehicle can be released into the blood by laser irradiation in the surgery treatment. Therefore, the development of structural deformation methods based on the IR-FEL against the amyloid fibrils will produce functional biomaterials in regenerative medicine and pharmaceutical fields. We obtained experimental and theoretical evidence on the processing of cellulose fiber by IR-FEL irradiation as well as amyloid fibrils [48,99]. The NEMD simulation showed that sugar chains are disassembled by laser irradiation tuned to 1360 cm^−1^ (C-O-H bending mode), and irradiation experiments using three wavelengths, 9.1 μm (νC-O), 7.2 μm (δC-O-H), and 3.5 μm (νC-H) showed that cellulose was disaggregated and the glycoside bonds are cleaved under atmospheric conditions. The IR-FEL can also be applicable for supplying low-molecular-weight sugars that can be employed for bioethanol production in the biorefinery process.

## 5. Conclusions

In this special issue, we presented a review of the dissociation of amyloid fibril induced by an intense infrared laser based on our combination studies of experiments and computer simulations. The IR-FEL can be oscillated through a linear accelerator and can excite chemical bonds’ vibrational modes, selectively. A structural feature of amyloid fibril is the β-sheet-stacking conformation, and it was revealed that the fibrous structure can be deformed to produce α-helix and other conformations during specific irradiation by the amide-I (6.1–6.2 μm) from the IR-FEL oscillation system. The experimental evidence was obtained by performing FT-IR spectroscopy, ThT binding assay, SEM, and X-ray scattering. The vibrational excitation of C=O stretching in the amide bonds is effective for cleavage of hydrogen bonds between the β-sheets, which was theoretically evidenced at both secondary structures and amino acid levels by using nonequilibrium and equilibrium molecular dynamics simulations. It can be believed that an intense infrared laser system based on IR-FEL should contribute to a degradation strategy for amyloid fibrils, although we should examine the effects on toxic oligomers on mammalian bodies towards the development of amyloidosis therapy. In addition, the IR-FEL irradiation system can be used as a processing tool for fibrous biomaterials constructed by β-sheet conformations in the medical and material engineering fields in the future.

## Figures and Tables

**Figure 1 ijms-24-03686-f001:**
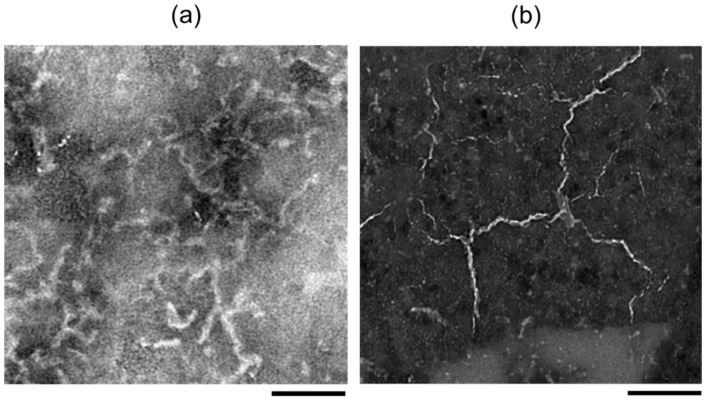
Transmission-electron microscopy images of lysozyme fibril (**a**) [21] and insulin fibril (**b**) [22]. Photographs were taken by negative staining using phosphotungstic acid. Black bar: 100 nm.

**Figure 2 ijms-24-03686-f002:**
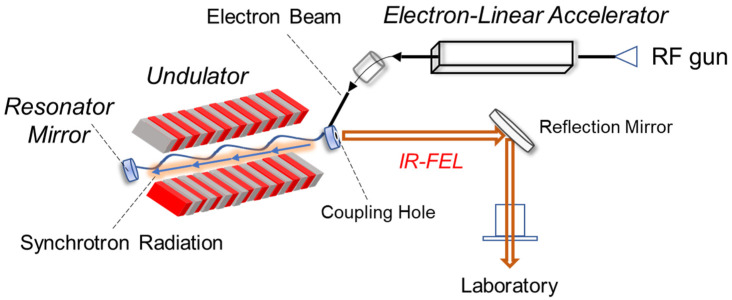
Outline of IR-FEL system. The oscillation system is mainly composed of electron-linear accelerator, undulator, and resonant mirrors. The undulator is formed by periodic N and S poles. The laser beam is transported from a coupling hole on the upstream mirror to user room via reflection mirrors set in a vacuum waveguide.

**Figure 3 ijms-24-03686-f003:**
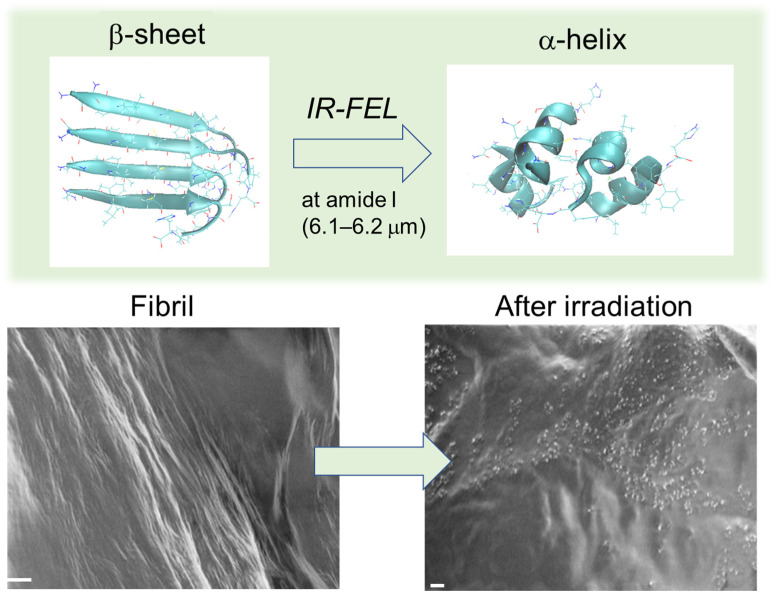
IR-FEL-mediated dissociation of amyloid fibril. Upper: conversion from β-sheet to α-helix in β2-microglobulin 11-residue peptide fibril [60]. Each conformation was displayed as a 3D model produced by equilibrium molecular dynamics simulation. Below: scanning electron microscopy (SEM) images of calcitonin 5-residue peptide fibril before and after IR-FEL irradiation [52]. White bar: 2 μm.

**Figure 4 ijms-24-03686-f004:**
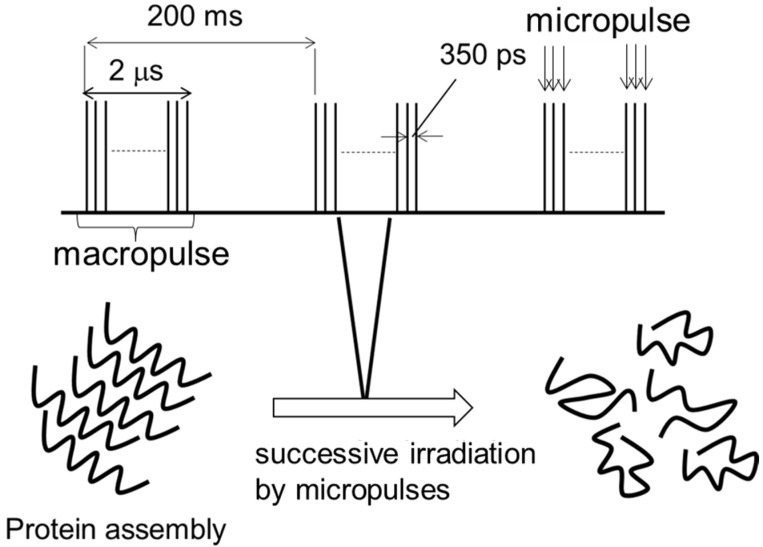
Disassembly of protein fibrils by successive irradiation of IR-FEL micropulses [58].

**Figure 5 ijms-24-03686-f005:**
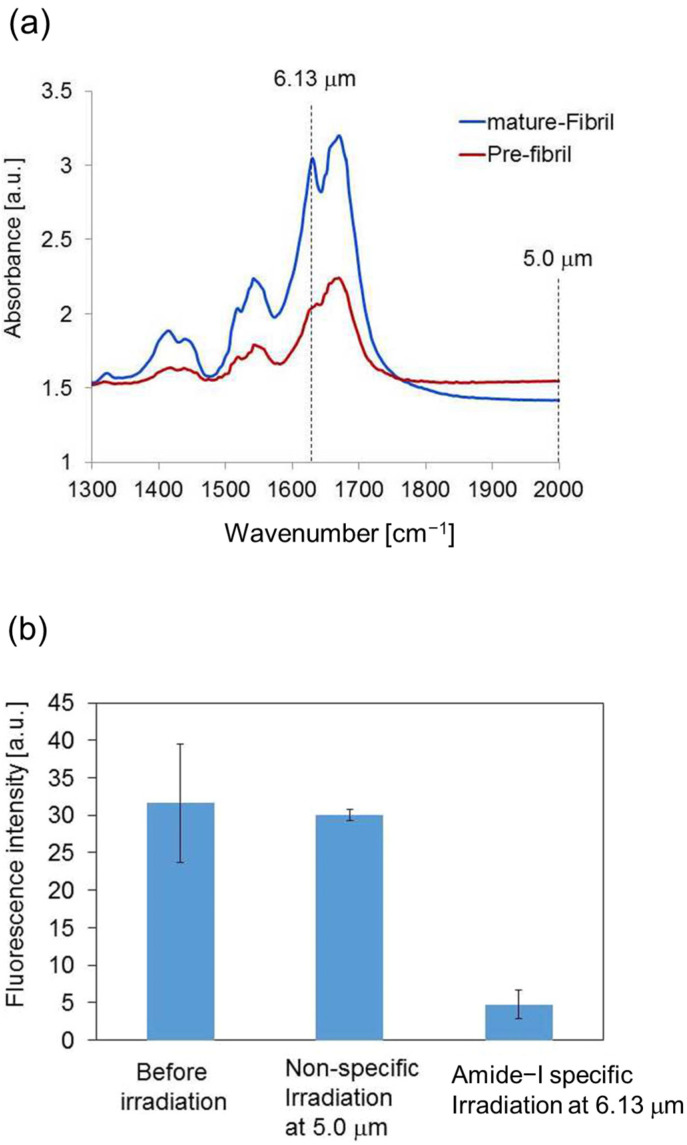
(**a**) Infrared spectra of the Pre-fibril (red) and mature-Fibril (blue) of the GNNQQNY peptide. The IR-FEL irradiation wavelengths of 6.13 μm and 5.0 μm used in this study were marked as dotted lines. (**b**) ThT binding assay. The reagent concentration is 25 μM and the laser energy is about 10 mJ per one macropulse at both wavelengths.

**Figure 6 ijms-24-03686-f006:**
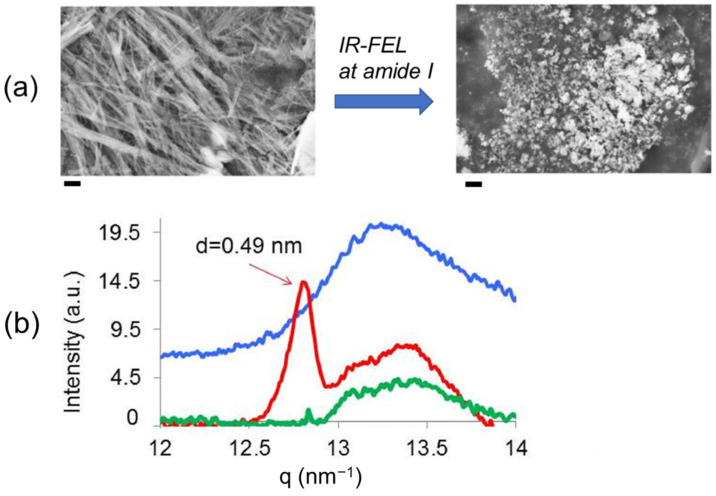
(**a**) SEM images of GNNQQNY peptide fibril before (left) and after (right) IR-FEL irradiation. Black bar: 200 nm. (**b**) WAXS spectra. Blue: Pre-fibril; red: mature-Fibril; green: irradiated fibril. Intermolecular distance (d) is equal with 2π/q.

**Figure 7 ijms-24-03686-f007:**
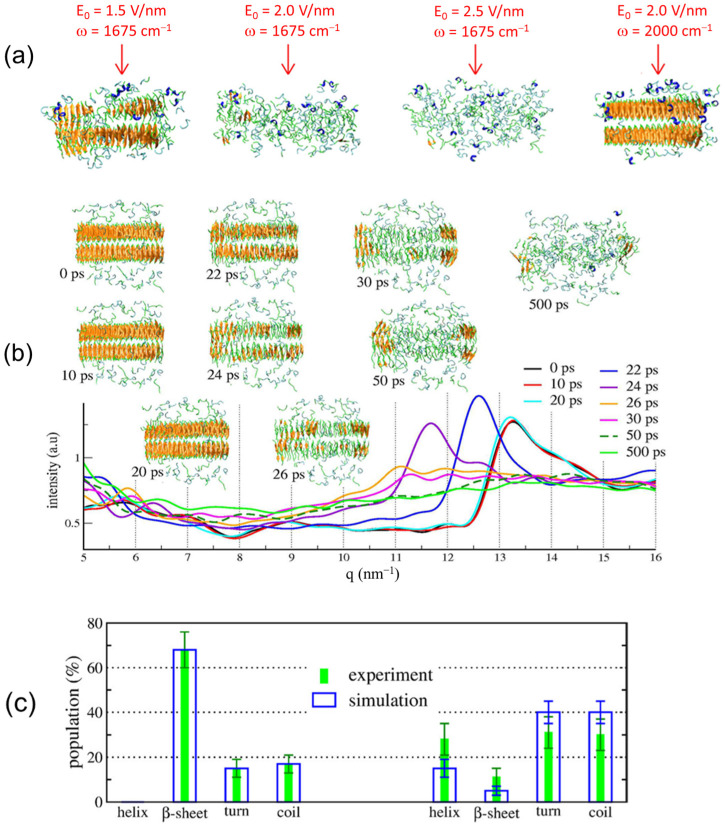
(**a**) The peptide assembly after 500 ps NEMD simulations at different laser intensities and frequencies. (**b**) Spectral changes in WAXS and the corresponding assemblies during NEMD simulation with E_0_ = 2.0 V/nm at 1675 cm^−1^. (**c**) Populations of protein secondary conformations in the peptide before (left) and after (right) laser irradiations at an experimental laser frequency of 1631 cm^−1^ (green) and at simulation laser frequency of 1675 cm^−1^ (blue).

**Figure 8 ijms-24-03686-f008:**
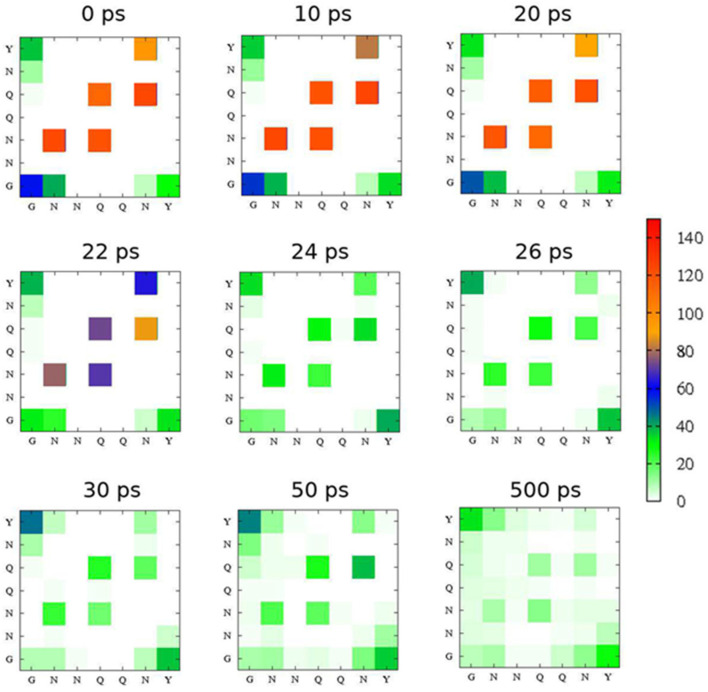
Intermolecular hydrogen-bond contact maps during the simulations with 2.0 V/nm at 1675 cm^−1^. The number of hydrogen-bond contacts are labeled by colors.

**Figure 9 ijms-24-03686-f009:**
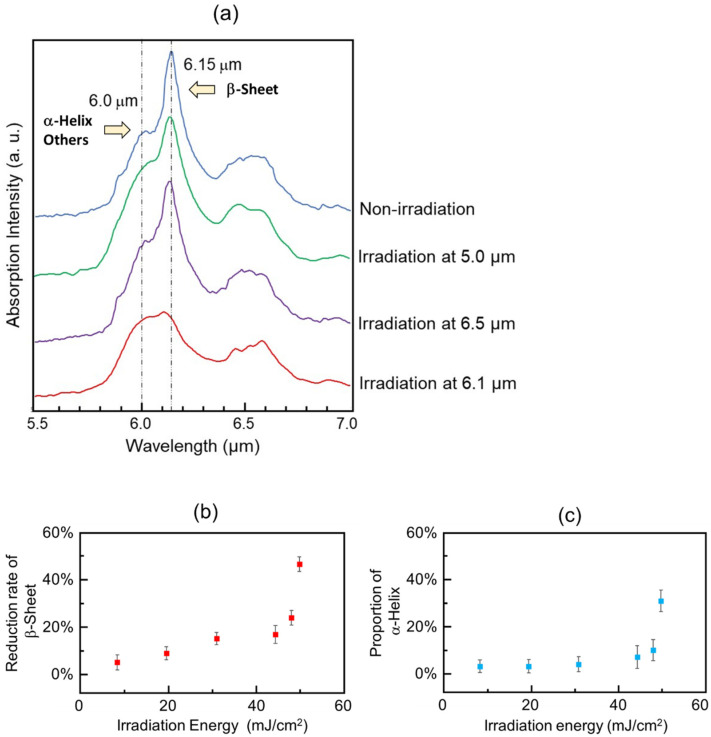
(**a**) Infrared microspectroscopy spectra of β2M peptide fibril before (blue) and after IR-FEL irradiations at 5.0 μm (green), 6.5 μm (violet), and 6.1 μm (red). (**b**) Irradiation-energy dependency of the reduction of the β-sheet. (**c**) Irradiation-energy dependency of the formation of the α-helix.

**Figure 10 ijms-24-03686-f010:**
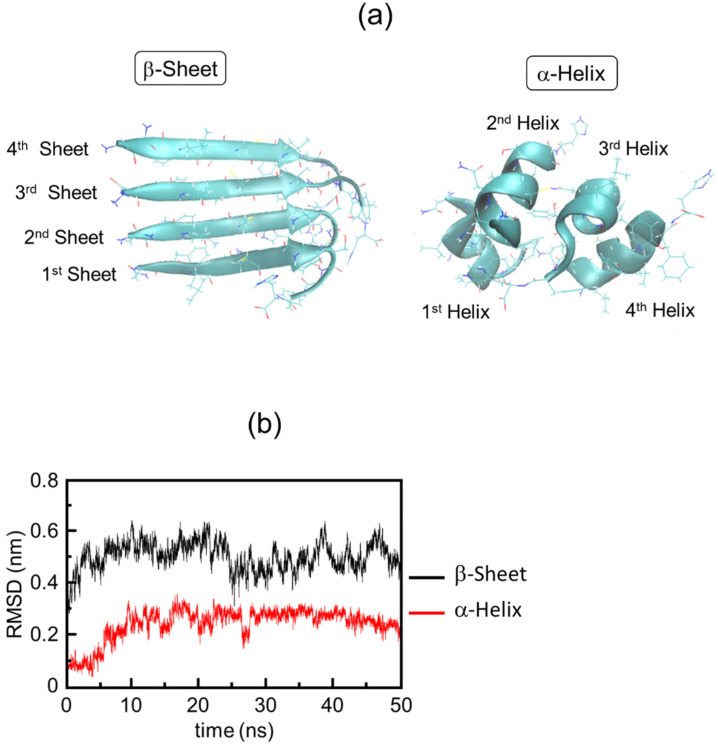
Equilibrium molecular dynamics simulation analysis. (**a**) β-Sheets (left) were extracted from the 11-residue peptide from β2M and the α-helix (right) was created by restricting the dihedral angle of the peptide main chain. (**b**) Time evolution of RMSD values (nm).

**Figure 11 ijms-24-03686-f011:**
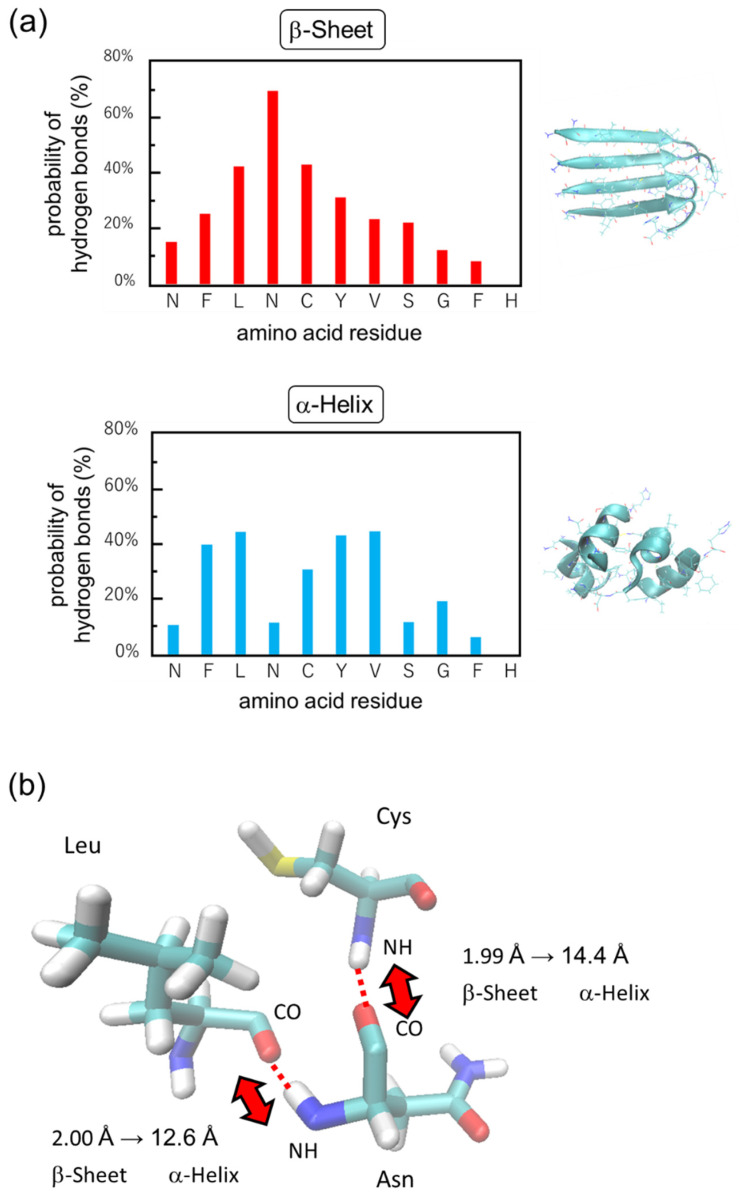
Molecular simulation of hydrogen bonds in 11-resiude peptide of β2M. (**a**) Probability of hydrogen bonds. (**b**) A 3D stick model of hydrogen bonds between N4 and L3 or C5. Light blue: carbon; red: oxygen; blue: nitrogen; white: hydrogen.

**Figure 12 ijms-24-03686-f012:**
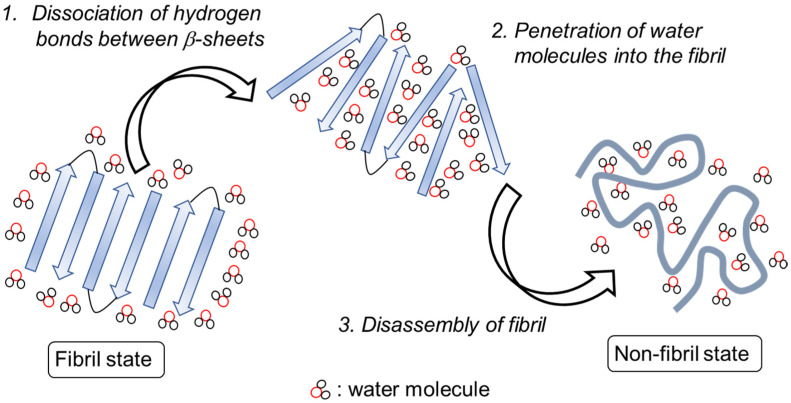
IR-FEL mediated disassembly of amyloid fibril. First, hydrogen bonds between the β-sheets are cleaved by vibrational excitation of amide bonds. Second, water molecules enter the core of the fibril and disturb the re-assembly of the peptide chains. Finally, the fibril is disassembled into the non-fibrous state containing α-helix and other conformations.

**Table 1 ijms-24-03686-t001:** Amyloidosis-related proteins and their amide I bands in fibril states.

Amyloidosis-Related Proteins	Amide I (cm^−1^)	Ref.
Aβ_1-40_/Aβ_1-42_ (Alzheimer’s disease)	1630/1626	[8,9]
Tau (Frontotemporal dementia)	1630	[10]
Polyglutamine (Huntington’s disease)	1625	[11]
Transthyretin (Amyloidotic polyneuropathy)	1615	[12]
Prion protein (Creutzfeldt-Jakob disease)	1626/1635	[13]
S100A6 (Amyotrophic lateral sclerosis)	1625	[14]
α-Synuclein (Parkinson’s disease)	1630	[15]
Lysozyme (Hereditary amyloidosis)	1614	[16]
Calcitonin (Thyroid medullary carcinoma)	1639	[17]
Insulin (Subcutaneous edema)	1628	[18]
β2-Microglobulin (Kidney dialysis amyloidosis)	1629	[19]

## Data Availability

All data in the experimental and theoretical studies are available from the corresponding authors (T.K. and P.H.N.) and our original papers.

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
