# Peer review of "Disassembly of Amyloid Fibril with Infrared Free Electron Laser"

_ijms, 2023, doi:10.3390/ijms24043686_

Round 1

Reviewer 1 Report

This manuscript describes adequately the application of the accelerator-induced infra-red free electron laser on the study of the disassembly of different types of amyloid fibrils and the conformational changes following their dissociation. It reviews the uses of several approaches including SEM, WAXS scattering and molecular dynamics simulations to track the effects of IR-FEL pulses on these amyloids. The manuscript is well-written with comprehensive details in recent studies in the use of IR-FEL. However, the authors seem to ignore very important details when discussing the effects of the IR-FEL on the dissociation of fibrils; for example i) the number of pulses required to induce such an effect (single or multiple pulses effect), ii) the pulse energy, iii) pulse duration, and iv) and what was the full width half maximum at the focal spot, and how does these factors affect the fibrils dissociation (see lines 148-184). 

Author Response

Dear Reviewer,

Thank you very much for reviewing our paper. We made responses to your comments as described below and revised the manuscript. We would appreciate it if you could review the responses and the revised paper.

Comment: This manuscript describes adequately the application of the accelerator-induced infra-red free electron laser on the study of the disassembly of different types of amyloid fibrils and the conformational changes following their dissociation. It reviews the uses of several approaches including SEM, WAXS scattering and molecular dynamics simulations to track the effects of IR-FEL pulses on these amyloids. The manuscript is well-written with comprehensive details in recent studies in the use of IR-FEL. However, the authors seem to ignore very important details when discussing the effects of the IR-FEL on the dissociation of fibrils; for example i) the number of pulses required to induce such an effect (single or multiple pulses effect), ii) the pulse energy, iii) pulse duration, and iv) and what was the full width half maximum at the focal spot, and how does these factors affect the fibrils dissociation (see lines 148-184).

Answer: Thank you so much for your positive comment and the valuable suggestion. We added details about the laser parameters such as the number of pulses, pulse energy, pulse duration, and the diameter of laser beam, and the effect of those parameters on the fibril dissociation was described in the revised version (line 132-138) as follows:

The laser beam from IR-FEL oscillation system was introduced to the surface of the fibril sample, and the beam diameter was focused to about 0.3-0.4 mm using barium fluoride lens. The pulse energy was 9−10 mJ, and the local destruction of the sample surface can be seen after 10 macropulses where the duration of one macropulse is 2 ms [74]. Below 5 mJ, the irradiation effect on the fibril dissociation was weak. In this study, the number of irradiation pulses were set to 900 macropulses (= 3 min) to induce full dissociation of the fibril state.

We added one more reference [74] that shows how is the minimum number of pulses for dissociation of the biomolecular aggregate.

We appreciate it if you could consider the above revision.

That’s all.

Sincerely yours

Takayasu Kawasaki, Ph.D.

Accelerator Laboratory, High Energy Accelerator Research Organization, 1-1 Oho, Tsukuba, Ibaraki 305-0801, Japan.

Phone: +81-29-864-5200-2014, Fax: +81-29-864-3182

E-mail: takayasu.kawasaki@kek.jp

Reviewer 2 Report

After reviewing this manuscript, I feel that this work is publishable. Yet, the authors need to consider the following points to further improve the quality of this manuscript.

(1)     The format and the writing style does not look like this work is a review type of work. It is like a research paper. Yet, the authors stated that “. In this review, we would like to introduce the IR-FEL oscillation system briefly and describe combination studies of experiments and 23…”.

(2)     In the introduction part, some references are not representative and were not published in high impact journals. For example, there is more suitable review work for the statement in line 44 for the IR assignment of amyloid fibril (refer to FTIR reveals structural differences between native beta-sheet proteins and amyloid fibrils, by FÄNDRICH) than the reference 20. There are similar problems for references 21-27. These references are not representative and high impact literature works.

(3)     In almost amyloid fibril? Or In most amyloid fibril? These errors are throughout this manuscript.

(4)     As for the presented Figures in the work, if this work is a review work, these figures should get copyright permissions.

(5)     The authors stated that free electron IR pulse can convert beta-sheet structure to alpha-helix structure. I have to say that for short peptide like GNNQQNY (7 residues) as well as other amyloidogenic peptides, it is known that they prefer to being at an so-called “intrinsically disordered” status. Please address this issue properly.

(6)     In the future outlook section, the authors stated that “Therefore, it can be supposed that targeting the binding of cholesterol in lipid membrane to the amyloid fibril may affect the progression of the pathogenicity, and a therapeutic approach based on this concept may lead to the next strategy for the treatment of AD.” It is difficult for the readers to see how free electron IR could be used in this application. I have to say that this perspective has nothing to do with this work.

Author Response

Dear Reviewer,

Thank you very much for reviewing our paper. We made responses to your comments as described below and revised the manuscript. We would appreciate it if you could review the responses and the revised paper.

Comment 1: The format and the writing style does not look like this work is a review type of work. It is like a research paper. Yet, the authors stated that “. In this review, we would like to introduce the IR-FEL oscillation system briefly and describe combination studies of experiments and 23…”.

Answer 1: Thank you so much for your comment. As you suggested, we described about our two representative papers in the second half in this review paper. Because we would like to emphasize that these joint studies of experiments and molecular simulations are very critical for showing the effect of IR-FEL on amyloid fibril dissociation. Nonetheless, we described more simply than the original papers in the revised version so as not to overlap with the previous phrases. We would appreciate it if you could consider this idea on our review paper in this special issue.

Comment 2: In the introduction part, some references are not representative and were not published in high impact journals. For example, there is more suitable review work for the statement in line 44 for the IR assignment of amyloid fibril (refer to FTIR reveals structural differences between native beta-sheet proteins and amyloid fibrils, by FÄNDRICH) than the reference 20. There are similar problems for references 21-27. These references are not representative and high impact literature works.

Answer 2: Thank you so much for your suggestion. We adopted “Fändrich M, Forge V, Buder K, Kittler M, Dobson CM, Diekmann S. Proc Natl Acad Sci U S A. 2003 Dec 23;100(26):15463-8. doi: 10.1073/pnas.0303758100.” for [20] and changed the references for [23]-[27] to more representative and higher impact papers as follows:

[23] Newberry, R.W.; Raines, R.T. A prevalent intraresidue hydrogen bond stabilizes proteins. Nat. Chem. Biol. 2016, 12, 1084-1088. doi: 10.1038/nchembio.2206

[24] Fitzpatrick, A.W.; Knowles, T.P.; Waudby, C.A.; Vendruscolo, M.; Dobson, C.M. Inversion of the balance between hydro-phobic and hydrogen bonding interactions in protein folding and aggregation. PLoS Comput. Biol. 2011, 7, e1002169. doi: 10.1371/journal.pcbi.1002169

[25] Close, W.; Neumann, M.; Schmidt, A.; Hora, M.; Annamalai, K.; Schmidt, M.; Reif, B.; Schmidt, V.; Grigorieff, N.; Fändrich, M. Physical basis of amyloid fibril polymorphism. Nat. Commun. 2018, 9, 699. doi: 10.1038/s41467-018-03164-5

[26] Fändrich, M.; Meinhardt, J.; Grigorieff, N. Structural polymorphism of Alzheimer Abeta and other amyloid fibrils. Prion. 2009, 3, 89-93. doi: 10.4161/pri.3.2.8859

[27] Tycko, R.; Wickner, R.B. Molecular structures of amyloid and prion fibrils: consensus versus controversy. Acc. Chem. Res. 2013, 46, 1487-96. doi: 10.1021/ar300282r

Comment 3: In almost amyloid fibril? Or In most amyloid fibril? These errors are throughout this manuscript.

Answer 3: Thank you so much for your suggestion. As you pointed out, whether “most” or “almost” may confuse the readers. As for line 21, we changed as follows: We have ever found that several different kinds of amyloid fibrils in amino acid sequences were commonly disassembled by the irradiation tuned to amide I (6.1-6.2 um) where abundance of b-sheet decreased while that of a-helix increased by the vibrational excitation of amide bonds.

As for line 50-51, we changed “almost” into “many” in line 45 of revised version.

Comment 4: As for the presented Figures in the work, if this work is a review work, these figures should get copyright permissions.

Answer 4: Thank you for your kind comment. We already checked the Right and permissions for each figure. For Fig. 4, we obtained permission from Springer Nature publisher, and as for Fig. 1, 3, and 5-11, we can use these figures freely because those figures are published under open access policy and creative commons licenses. Table 1, Fig. 2, and Fig. 12 are original in this paper.

Comment 5: The authors stated that free electron IR pulse can convert beta-sheet structure to alpha-helix structure. I have to say that for short peptide like GNNQQNY (7 residues) as well as other amyloidogenic peptides, it is known that they prefer to being at an so-called “intrinsically disordered” status. Please address this issue properly.

Answer 5: Thank you so much for your valuable suggestion. As you suggested, it is known that several proteins such as Sup35 in bacteria contain intrinsically disordered regions which tend to form b-sheet domains in crowed milieu. Also, it is suggested that the intrinsically disordered proteins work functionally in cells. Those proteins are so called as functional amyloids. We have not yet investigated the effect of IR-FEL on those natively disordered regions in vivo, although we found that the b-sheet rich assembly was dissociated to the non-fibrous conformation in the peptide fibrils in vitro. However, there is a possibility that the cell function related with the functional amyloids can be altered by the physical engineering technique using IR-FEL. How the IR-FEL gives the effect on the natively disordered proteins in the living organism is very interesting for us, and the study on the effect of IR-FEL on the cell functions related with intrinsically disordered peptides should be a next subject. We added comments and references [91,92] on this issue in the revised version in line 316-321. Thank you very much for your suggestion.

Comment 6: In the future outlook section, the authors stated that “Therefore, it can be supposed that targeting the binding of cholesterol in lipid membrane to the amyloid fibril may affect the progression of the pathogenicity, and a therapeutic approach based on this concept may lead to the next strategy for the treatment of AD.” It is difficult for the readers to see how free electron IR could be used in this application. I have to say that this perspective has nothing to do with this work.

Answer 6: Thank you for your valuable suggestion. A study on decomposing cholesterol by using IR-FEL was reported by other group [85], and we added a possible use of IR-FEL for targeting lipid membrane containing cholesterol in this discussion part (line 283-292) as follows: In recent years, it has been suggested that the binding of Aβ to cell membranes is critical step for the onset of AD pathogenicity and hydrophobic interactions of cholesterol molecule with Ab plays important roles for the b-sheet oligomerization [64,81-84]. Therefore, it can be supposed that targeting the binding of cholesterol in lipid membrane to the Ab fibril may affect the progression of the pathogenicity. The IR-FEL can decompose the cholesterol in blood vessels by tuning the irradiation wavelength to the ester bonds [85]. It can be supposed that the Ab binding to the lipid membrane can be perturbed by excluding the cholesterol by the laser irradiation. In future, a therapeutic approach based on this concept may lead to the novel strategy for the treatment of AD.

We appreciate it if you could consider the above revision.

That’s all.

Sincerely yours

Takayasu Kawasaki, Ph.D.

Accelerator Laboratory, High Energy Accelerator Research Organization, 1-1 Oho, Tsukuba, Ibaraki 305-0801, Japan.

Phone: +81-29-864-5200-2014, Fax: +81-29-864-3182

E-mail: takayasu.kawasaki@kek.jp
